# Findings and Challenges in Replacing Traditional Uterine Cervical Cancer Diagnosis with Molecular Tools in Private Gynecological Practice in Mexico

**DOI:** 10.3390/v16060887

**Published:** 2024-05-31

**Authors:** José L. Castrillo-Diez, Carolina Rivera-Santiago, Silvia M. Ávila-Flores, Silvia A. Barrera-Barrera, Hugo A. Barrera-Saldaña

**Affiliations:** 1Leire Genomic Laboratories, León 37000, Mexico; 2Columbia Biotec, Columbia Laboratories, Tlalpan 14090, Mexico; crivera@gcolumbia.com; 3Columbia Laboratories, Basic Scientific Research Division, Mexico City 04000, Mexico; 4Innbiogem SC/Vitagenesis SA at National Laboratory for Services of Research, Development, and Innovation for the Pharma and Biotech Industries (LANSEIDI) of CONACyT Vitaxentrum Group, Monterrey 64630, Mexico; silviaabb92@gmail.com; 5Facultades de Medicina y Ciencias Biológicas, Universidad Autónoma de Nuevo León, San Nicolás de los Garza 66455, Mexico

**Keywords:** uterine cervical cancer screening, HPV genotyping, p16/Ki67 biomarkers, gynecology private practice

## Abstract

We have been encouraging practicing gynecologists to adopt molecular diagnostics tests, PCR, and cancer biomarkers, as alternatives enabled by these platforms, to traditional Papanicolaou and colposcopy tests, respectively. An aliquot of liquid-based cytology was used for the molecular test [high-risk HPV types, (HR HPV)], another for the PAP test, and one more for p16/Ki67 dual-stain cytology. A total of 4499 laboratory samples were evaluated, and we found that 25.1% of low-grade samples and 47.9% of high-grade samples after PAP testing had a negative HR HPV-PCR result. In those cases, reported as Pap-negative, 22.1% had a positive HR HPV-PCR result. Dual staining with p16/Ki67 biomarkers in samples was positive for HR HPV, and 31.7% were also positive for these markers. Out of the PCR results that were positive for any of these HR HPV subtypes, n 68.3%, we did not find evidence for the presence of cancerous cells, highlighting the importance of performing dual staining with p16/Ki67 after PCR to avoid unnecessary colposcopies. The encountered challenges are a deep-rooted social reluctance in Mexico to abandon traditional Pap smears and the opinion of many specialists. Therefore, we still believe that colposcopy continues to be a preferred procedure over the dual-staining protocol.

## 1. Introduction

The specific detection of the human papillomavirus (HPV) subtype has favored the reduction in the mortality and morbidity rate in women due to cervical cancer (CC) worldwide [1]. This reduction has been achieved due to the widespread use of liquid-based cytology preparations to obtain cervical samples and the implementation of new molecular technologies for HPV genotyping by polymerase chain reaction (PCR) [2,3,4,5].

Globally, CC is the fourth most common cancer in women, with 604,000 new cases in 2020 [5]. About 90% of the 342,000 deaths caused by CC occurred in low- and middle-income countries [6,7]. The highest rates of this tumor incidence and mortality are in sub-Saharan Africa (SSA), Central America, and South-East Asia. Regional differences in the CC burden are related to inequalities in access to vaccination, screening and treatment services, risk factors (including human immunodeficiency virus (HIV) prevalence and social and economic determinants such as sex), gender biases, and poverty. Women living with HIV are six times more likely to develop CC compared to the general population, with an estimated 5% of all cases attributable to this infection. The contribution of HIV to CC disproportionately affects younger women, and as a result, 20% of children who lose their mother to cancer do so due to this neoplasia [8].

In Mexico, this neoplasm represents the second cause of cancer in women, with 9440 new cases per year, and the second cause of death, with 4340 cases [9]. Among women with invasive CC, around 70% are diagnosed with locally advanced disease, which highlights deficiencies in timely diagnosis [9]. All these data reinforce the need to continue implementing early screening strategies for CC in Mexico and in other Latin American countries with similar incidences. 

In recent years, the diagnosis of HPV infections by detection of viral DNA and PCR genotyping of high-risk (HR) variants in cervical samples has replaced the traditional Pap smear due to its higher sensitivity [1,10]. The improvement in the HR HPV detection capacity offered by PCR and its automation has led various countries, such as Australia, to abandon the Papanicolaou test as a CC screening tool [11]. On the other hand, in Latin America, the efforts to implement HPV genotyping by PCR continue to be limited, and greater awareness of this serious female health problem is necessary in public institutions and private hospitals [12].

HPV is a highly transmissible virus that leads to transient infections, with several factors increasing the risk of its persistence, including genetics, age, smoking, and the specific genomic sequence of the infecting virus [13]. To date, more than 150 subtypes of papillomavirus viruses have been described that, based on their association with CC, have been classified as very HR (HPV-16 and HPV-18), twelve HR subtypes (HR12), and other low-risk (LR) types that are associated with benign mucosal lesions [14]. The high- and very-HR HPV subtypes (16, 18, and HR12) are associated with malignant lesions and cause approximately 70% of CC cases worldwide [15,16].

The cytological course caused by transient HPV infection begins with low-grade squamous intraepithelial lesions, of which 90% revert to healthy epithelium. However, HR-HPV-specific infections aggravated by host risk factors are more difficult to reverse, and their persistence leads to high-grade squamous intraepithelial lesions (moderate or high dysplasia) that can progress to CC [15]. Cytological assessment by the expert cytotechnologist is highly relevant for predicting the course of HPV infection. However, it has its limitations, such as low sensitivity and the impossibility of distinguishing persistent infection from reinfection [17]. These are examples in which molecular tools such as PCR, given their greater sensitivity and ability to identify the subtype of HPV infection, outperform traditional diagnostics [4,5].

To date, seven PCR assays for the detection and genotyping of HPV from cervical cell samples have been validated. Three of them are tests aimed at amplifying a region of the L1 gene (Abbott Real-Time HR HPV Test, Anyplex II HPV HR Detection, and Cobas 4800 HPV Test). At the same time, another four are assays that amplify early-region genes (BD Onclarity HPV Assay, HPV-Risk Assay, PapilloCheck HPV-Screening Test, and Xpert HPV) [18].

In Mexico and the rest of Latin America, strategies have been developed to implement these molecular PCR tests for HPV detection in public and private institutions [12]. However, automated PCR methods, such as those used in our laboratories (Cobas 4800 HPV Test), stand out from other methods for their ability to process many simultaneous samples and their speed and diagnostic accuracy [19]. Using an internal control for co-amplification of a human gene makes it possible to practically eliminate the analysis of invalid samples and the presence of false negatives [19].

Although the specific detection of HPV genotyping by PCR is currently the “gold-standard” technique in the early diagnosis of CC, complementary technologies have been developed for diagnosing premalignant cervical lesions in liquid cytology samples [20]. The p16INK4a (p16) protein is a regulatory protein of the cell cycle under normal physiological conditions [21]. This biomarker is effective in histological samples and is widely used to improve the reproducibility of cervical biopsy assessment and accuracy in detecting premalignant lesions [20,22,23,24]. Likewise, the simultaneous detection of p16 and Ki67 (a proliferation biomarker) within the same cervical epithelial cell has been proposed as a marker of cellular transformation mediated by infections with the 12 HR HPV genotypes (HR12) [25]. This combination of biomarkers (p16/Ki67 dual-stain cytology, Cintec-Plus) has provided excellent results in cervical cytological samples where it has been used for the detection of premalignant and malignant lesions of CC [26,27,28,29,30]. 

Given the proven advantages of the herein-described molecular tests, we have been implementing them in our laboratories and boosting their adoption among private practicing gynecologists in Mexico.

## 2. Materials and Methods

### 2.1. Clinical Samples and Diagnostic Algorithms

A total of 4499 cervix samples received consecutively in our laboratories were analyzed using three strategies: (a) molecular PCR analyses (to screen for the presence of HPV infections); (b) liquid-based cytology to search for cellular alterations suggestive of HPV infections, and (c) finally, if the medical expert followed the triage, they requested the laboratory to perform dual-staining cytology (to assess if the cellular transformation process has already started). PCR was used to test all 4499 samples, 3806 were subjected to liquid-based cytology, and 567 samples were analyzed by p16/Ki67 dual-stain cytology.

### 2.2. Clinical Specimen Sampling

Cervical samples from the cervix were taken by gynecologists in private clinical practice using a cervix brush and deposited in a transport medium (ThinPrep or Roche Cell Collection Medium). The vials with the samples were sent at room temperature to the laboratory, where they were stored and refrigerated (4 degrees Celsius) until processing.

### 2.3. Liquid-Based Cytology (PAP Test)

Liquid-based cytology slides were prepared by a cytotechnologist and interpreted according to the Bethesda System for Reporting Cervical Cytology (third edition, 2017). In contrast, a pathologist reviewed 50% of negative samples and 100% of the positive ones for quality control and quality assurance.

### 2.4. HPV PCR Assay

The COBAS 4800 HPV Test (Roche) is an FDA-approved and validated qualitative test device for detecting HPV DNA in swabs from the cervical canal. This test amplifies target DNA isolated from cervical epithelium by real-time PCR to detect HPV 16 and HPV 18, along with a simultaneous pooled result for 12 other HR genotypes in a single test. The entire procedure is automated, and the manufacturer’s instructions are followed. The COBAS 4800 HVPV Test Primers are used to amplify DNA from 14 HR-HPV types (16, 18, 31, 33, 35, 39, 45, 51, 52, 56, 58, 59, 66, and 68) in a single analysis, where probes with four different reporter dyes screen different targets in the multiplex reaction: dye 1 screens 12 pooled HR-HPVs (31, 33, 35, 39, 45, 51, 52, 56, 58, 59, 66, and 68), dyes 2 and 3 screen for HPV 16 and 18, respectively, while dye 4 targets the human β-globin gene to provide a control for uterine cervical cell adequacy for extraction and amplification. 

### 2.5. p16/Ki67 Dual-Stain Cytology

One slide for each sample was prepared for PAP testing using a Cytospin chamber adapter and subjected to p16/Ki67 dual-stain cytology using the CINtec Plus Cytology Kit (Roche Laboratories, Indianapolis, USA), according to the manufacturer’s instructions. Immunohistochemistry staining was performed using a BenchMark GX Stainer followed by evaluation by a trained cytotechnologist. An initial evaluation was performed to confirm the presence of the minimal criteria for squamous cellularity defined by the Bethesda terminology. Subsequently, the slide was checked for the presence of double-immunoreactive cervical epithelial cells, that is, cells with simultaneous brown cytoplasmic p16 immunostaining and Ki67 red nuclear immunostaining, which were interpreted as positive by double-stained cytological analysis regardless of the morphological interpretation. A pathologist reviewed all cases with positive cells for double-staining to confirm the result. 

The PCR results were compared with the PAP findings using a contingency table and chi-square test. All data analyses were performed using Graph Pad Prism 10 (GraphPad Software, Inc. (Boston, MA, USA). 

## 3. Results

Four thousand and four hundred ninety-nine liquid-based cervical samples were performed on samples from patients referred by private gynecologists mostly from central regions of Mexico. A molecular PCR study for genotyping of HPV was performed on all samples to detect the 14 most common HR genotypes of HPV to contribute to the early prevention of CC. In parallel, Pap tests were performed on 84.6% (*n* = 3806) of samples.

The global distribution of the results of the viral genotypes revealed by the PCR analysis is described in Figure 1. In 99.4% of cases, the samples received were considered adequate for this test, given that they rendered positive during the amplification of the internal control gene, i.e., the genomic β-globin gene. In cases where samples did not pass this control (0.6%), a second sample was requested from the gynecologist. This request was met on 53.8% of occasions (14/26). Thus, the final number of PCR results was 4487, or 99.7% of the samples received. 

Out of the 4487 results obtained, 68% tested negative for any of the 14 HR HPV genotypes analyzed. Out of the 1438 samples that tested positive for HR HPV (32%), 1261 (87.7%) were positive for any of the 12 HR HPV subtypes present in the group. In 12.3% of cases, the positive result corresponded to HPV16 and/or HPV18 subtypes, considered very HR for CC. It was observed that 86% of positive cases presented an isolated viral genotype without involving genotypes 16 or 18, while 14% showed co-infections with one of the two most oncogenic HR HPVs (viral genotype 16 or 18). Most of the positive samples yielded positive PCR results for the group of 12 HR subtypes (HR12). Overall, 24.5% of patients had PCR results involving very HR viruses (HPV16 and/or HPV18), alone or in combination with HR12 viral types. The results of PCR viral genotyping distributed by age range are shown in Table 1.

The data shown in Table 1 correspond to 3826 samples for which data on the age of the patients were available. However, if the results are stratified by age ranges, the population under 25 years of age was the one that showed the highest percentage of overall positive HPV-PCR results: 44% (181/411) tested positive for one of the high or very HR subtypes. 

If the results distributed by viral subtype are analyzed, the population between 25 and 35 years of age always shows a higher risk of infection with high or very HR genotypes. More than 50% of the results were positive in this age range for any viral genotype alone or in co-infection: 50.1% (450/898) for HR12, 51.8% (57/110) for HPV16, 55.2% (16/29) for HPV18, and 54.5% (91/167) for a viral co-infection. 

If the different positive subtypes are analyzed by age range, the HR12 subgroup always shows the highest percentage: 84% (152/181) in patients younger than 25 years, 73.3% (450/614) for the age range of 25 to 35 years, and 72.4% (296/409) for those over 35 years. However, for the other subtypes, their behavior by age is vastly different. For HPV16, 5% (9/181) was obtained for those under 25 years of age, 9.3% (57/614) for the range of 25 to 35 years, and 10.8% (44/409) for others over 35 years of age. For HPV18, the data were 0.6% (1/181) for those under 25 years of age, 2.6% (16/614) for those 25 to 35 years of age, and 2.9% (12/409) for those over 35 years of age. The data for viral coinfections showed that 10.5% (19/181) correspond to those under 25 years of age, 14.8% (91/614) for the age range of 25 to 35 years, and 13.9% (57/409) for those over 35 years of age. 

Analyzing the results globally, it stands out that: (a) The population under 25 years of age presented a higher percentage of positive results for HR12 viral types (84%), compared to 77.3% of the group between 25 and 35 years of age and a figure of 72.4% for those over 35 years of age. (b) The population between 25 and 35 years of age presented a higher percentage of results for co-infection, with 14.8%, compared to 13.9% in the group over 35 years of age and 10.5% in those under 25 years of age. (c) The population over 35 years of age presented a higher percentage of positive results for HPV16 with 10.8%, compared to 9.3% for the age range of 25 to 35 and 5% for those under 25. Likewise, the population over 35 years of age presented a higher percentage of positive results for HPV18 (2.9%), compared to 2.6% for the age range of 25 to 35 and 0.6% for those under 25 years of age.

The global distribution of the 3806 Pap test results, which represent 84.6% of the total of 4499 PCR tests, is shown in Figure 2. The positive results were sub-classified as low grade (69.1%) or high grade (30.9%). All negative and positive samples from the PAP test underwent PCR testing for the detection of high or very high-risk (HR) HPVs. Using a contingency table, we compared the results of the gold standard for HPV detection (PCR) with those of the PAP test. Within these findings, we noted that the percentage of false positives in cytology was 5.6% (*n* = 200) of the total samples assessed. Additionally, Figure 2 presents the results of HR HPV-PCR among the 3184 samples identified as negative in the PAP test. Notably, among these, 703 samples (22.1%) yielded positive results for viral genotypes of high or very HR for CC. Within this subset of false negatives from cytology for HPV detection, 85.9% (*n* = 604) corresponded to the 12 HR subtypes (HR12), while 14.1% (*n* = 99) were attributed to HPV16 or HPV-18, which are very HR genotypes.

Table 2 shows the distribution by age of the 622 samples with a positive result for PAP, and whether they are low- or high-grade. The most striking results correspond to patients over 35 years of age: 110 samples (45 low-risk and 65 HR) produced negative results in the PCR test, representing 45.6% of potential false positives (110/241). The percentages of the lack of correlation between PAP and HPV-PCR were lower in the other age ranges: 25.4% in the age range of 25 to 35 years [(53 + 21)/291] and 17.7% for those under 25 years [(10 + 6)/90].

Table 3 shows the HR HPV-PCR genotyping results in the 422 double-positive samples in both the HPV-PCR and PAP tests (*n* = 322 low-grade and *n* = 100 high-grade). A total of 61.8% of the samples were positive for HR12, 16% were positive for HR12 plus HPV16, and 15.2% were positive for isolated HPV16. Overall, 79.6% of the analyses obtained an isolated infection [(261 + 64 + 11)/422], and 20.4% corresponded to double or triple co-infections [(4 + 67 + 1 + 41)/422].

Figure 3 shows the results of the confirmation tests for CC performed using p16/Ki67 dual-stain cytology (CINTEC PLUS). A total of 536 samples (37.3%) from the subgroup of 1438 positive results for 14 HR HPV types were tested. A total of 31.7% were positive for dual staining. These results also confirm that 68.3% of the positive results for any of the HR HPV subtypes do not present cellular transformation, which highlights the importance of performing p16/Ki67 dual-stain cytology after PCR to avoid unnecessary colposcopies in the screening of CC patients.

These samples came from private practice gynecologists in private hospitals. In 100% of the cases, the gynecologists continued to perform the traditional Pap test in parallel with HPV genotyping analysis using PCR (Cobas). Although this double screening for the early diagnosis of CC represents an extra cost for their patients, gynecologists argue that the traditional Pap test allows them to identify other gynecological lesions (suspected infections by bacteria, fungi, or other viruses) that the PCR test alone does not allow. In Figure 4, we present a strategy to encourage the increased adoption of molecular methods, highlighting the benefits of employing these tools throughout the entire process, from sample collection to patient reporting. In our experience, gynecologists are aware of the good sensitivity and specificity of the molecular PCR test and the costs of the study for their patients are the only reason for not requesting the HPV genotyping test together with the traditional PAP test. 

Regarding the possible reasons why the dual-stain test is not widely requested within the global detection program for CC in the private gynecology sector, we can include the following: (1) Extra costs for many of the patients, for whom the cost of the PCR study already represents a large increase over the expense of the traditional PAP; (2) Ignorance on the part of the patients of the objective and scope of the staining in CC screening, which implies a longer explanation time in private medical consultations; and (3) Reticence from a subgroup of gynecologists (colposcopists) since the positive/negative result of the dual-stain test determines (or relativizes) the need for surgical intervention (colposcopy) in their patients. This result seems to interfere with their area of expertise and business, which may be one of the causes of the low return rate in the confirmatory dual-stain test.

## 4. Discussion

The results presented here correspond to 4499 samples received in our laboratories during three years for CC screening. The distribution of HPV subpopulations in all samples is comparable to other results published internationally. However, this study shows for the first time the results in the Mexican population within the private medicine market [1,9].

It should be noted that if we analyze the results within each age range, the population under 25 years of age was the one that showed the highest percentage of overall positive HR HPV-PCR results: 44% tested positive for one of the risk subtypes, high or very high, compared to 35.6% of positive results in the population between 25 and 35 years of age and 24.2% of positive results in the population over 35 years of age.

Likewise, our results make it possible to highlight that the population between 25 and 35 years of age presented a higher percentage of results for viral co-infections with high or very HR for CC viral types at 14.8% compared to 13.9% in the group over 35 years of age and 10.5% in those under 25 years of age. On the other hand, the population over 35 years of age was the one that presented a higher percentage of positive results for HPV16 and HPV18.

Once we organized the PCR results into a contingency table alongside the PAP results, we identified 5.6% false positives and 18.5% false negatives for the PAP results. These findings strongly support the advantages of using PCR as the standard test for detecting the human papillomavirus and preventing CC. The most striking results correspond to patients older than 35 years, who show 45.6% potential false positives. The percentages of PAP vs. HR HPV-PCR non-correlation were lower in the other age ranges: 25.4% in the age range of 25 to 35 years and 17.7% for those younger than 25 years. These results may suggest the existence of an age bias when interpreting PAP results in the older population, who are theoretically more prone to presenting with CC.

Likewise, our data allowed us to detect a percentage of 68.3% negative results using confirmatory p16/Ki67 dual-stain cytology in the population of positive patients in the HR HPV group. These results made it possible to reassure the patients who had initially received a worrying result from the PCR test and to reduce the performance of unnecessary invasive tests (for example, colposcopy).

Although the PCR test for HPV screening is well known by gynecologists, p16/Ki67 dual-stain cytology is not a widely requested test for CC screening. Out of the total HR-HPV-positive samples, only 37.3% were claimed for re-analysis with confirmatory dual staining. PCR tests detect molecular changes within the cell that are not manifested in morphological changes. This premise supports the promotion of PCR and p16/Ki67 dual-stain cytology as primary screening methods for HPV detection and CC prevention, as they provide greater diagnostic sensitivity to the physician. It is worth noting that, although there are other sensitive methods for HPV detection, such as NGS (Next-Generation Sequencing), this option is costly and would take longer to obtain results (even weeks). In addition, this method is not validated for the clinical diagnosis of CC.

We acknowledge the limitations of our study in enhancing the robustness of our results. One limitation is the absence of clinical information or pathological tissue analysis of the patients, which is crucial for determining the sensitivity and specificity parameters, particularly within our study and the evaluated population sample. However, the most important challenges encountered after carrying out these works are: (1) Continue promoting, among gynecologists, that the papillomavirus PCR technique (HR HPV-PCR) is the best method for CC screening due to the lower presence of false negatives and false positives. (2) Stimulate the study of dual staining in HR12-positive patients to focus colposcopy procedures only on those patients who have cancer cells. (3) Promote the establishment of new protocols agreed between the Medical Societies of specialists (Gynecology and/or Colposcopy) for the better diagnosis and treatment of patients with CC [31,32]. (4) Recently, a new global WHO strategy to achieve the elimination of CC as a public health problem by 2030 has been launched. It is called “Strategy 90-70-90”: 90% of girls (and boys in countries where resources allow) are to be fully vaccinated with the HPV vaccine by age 15, 70% of people are to be screened with a high-performance test, and 90% of persons identified with cervical disease will have received treatment [33]. However, there are large inequities still to be resolved. Significant disparities continue to increase between countries and within different regions of the same country. Large differences persist between sociodemographic groups, household income, and children’s access to health insurance. The aspects in which we should improve to achieve these objectives are: (a) Accelerate the implementation of HPV screening programs, (b) Take advantage of diagnostic and therapeutic innovations, and (c) Focus on equity [33]. 

To address these challenges, we are considering and proposing to (1) Disseminate in gynecology congresses in our countries the benefits of molecular diagnosis of HR HPV by PCR, insisting on its greater speed, specificity, and sensitivity. (2) Carry out dissemination campaigns of these new technologies in social networks aimed at the neediest female population in the prevention of CC. (3) Inform and achieve scientific discussion in specialized societies on the clinical risks associated with the unnecessary performance of many colposcopy procedures, for example, infertility or miscarriages. (4) Implement two innovations in our countries: (a) Promote self-collection programs for HR HPV-PCR screening, and (b) Test and validate new detection procedures close to the patient, such as point-of-care (POC) testing. The self-collection programs have already been validated by many countries around the world [34]. The sensitivity and specificity of self-sample HPV tests are like provider-collected HPV tests, and some devices are excellently accepted by women from very different countries [34,35,36]. Regarding the accessibility of less expensive and convenient methodologies than automated laboratory machines, especially for developing countries, there is great interest in POC testing instruments that could be fast, low-cost, and with a minimal training requirement [33].

## 5. Conclusions

The HR HPV genotyping results by PCR show the low specificity and selectivity of the PAP test for CC screening. Up to 47.9% false positives have been observed in samples with a cytology diagnosed as high-grade positive. Likewise, 22.1% of false negatives have been observed in samples diagnosed as negative in the PAP test. The results of p16/Ki67 dual-stain cytology in positive PCR samples for any of the HR HPV subtypes show that 68.3% do not present with cancer cells, highlighting the importance of performing these tests to avoid unnecessary colposcopies after the screening of possible patients with CC. 

The results of our interactions with practicing gynecologists can be summarized as follows. There is a deeply embedded societal reluctance to abandon traditional PAP tests given the long history of public relations campaigns to convince women and their doctors of their value as the preferred prevention tool for the diagnosis of CC. In the opinion of some specialists, colposcopy remains preferred to dual-staining protocols despite the invasive and frequently unnecessary practice. 

Eradicating CC still is a challenge in Mexico due to poor prevention education and the inefficacy of current PAP technology. Our goal is to educate doctors and their patients about the benefits of the new molecular screening technologies for effective primary screening for HPV and the diagnosis of CC. 

## Figures and Tables

**Figure 1 viruses-16-00887-f001:**
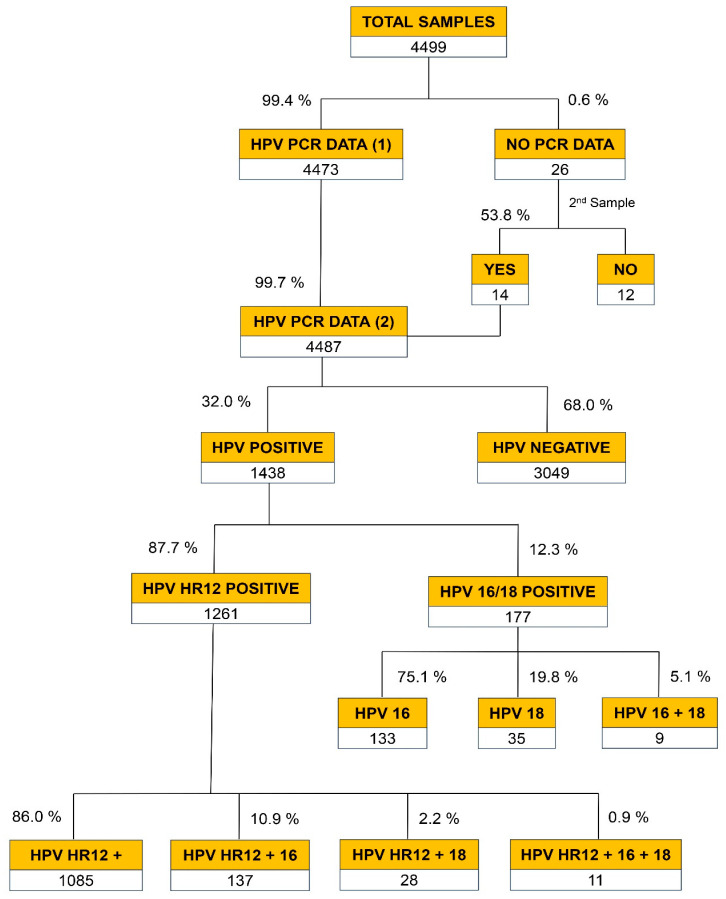
General distribution of HPV genotyping results by PCR. Findings are presented by type of virus: HPV-16 (16), HPV-18 (18), and a pool of other 12 HR-HPVs responsible for most cases of CC (HR12).

**Figure 2 viruses-16-00887-f002:**
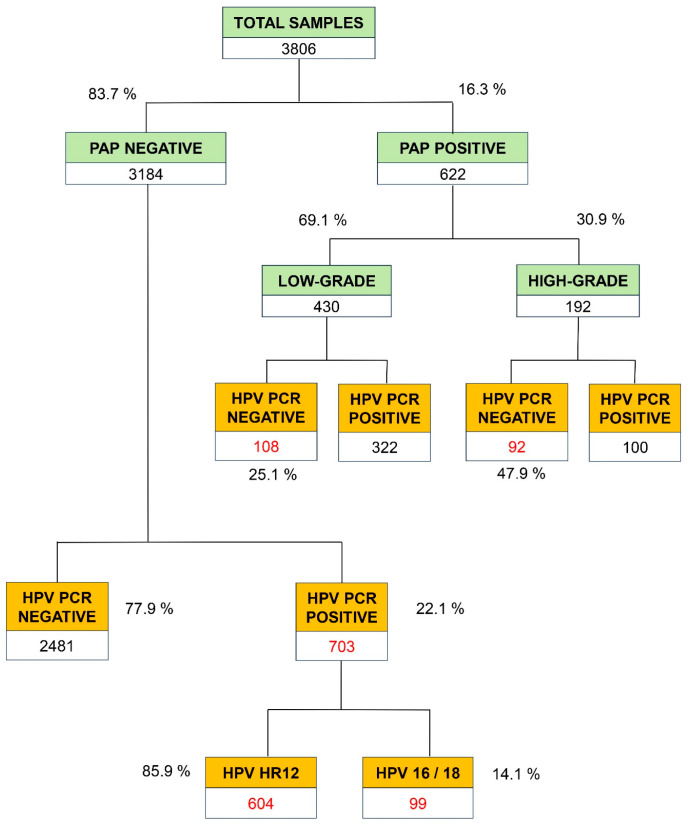
General distribution of PAP screening and HR HPV genotyping results by PCR.

**Figure 3 viruses-16-00887-f003:**
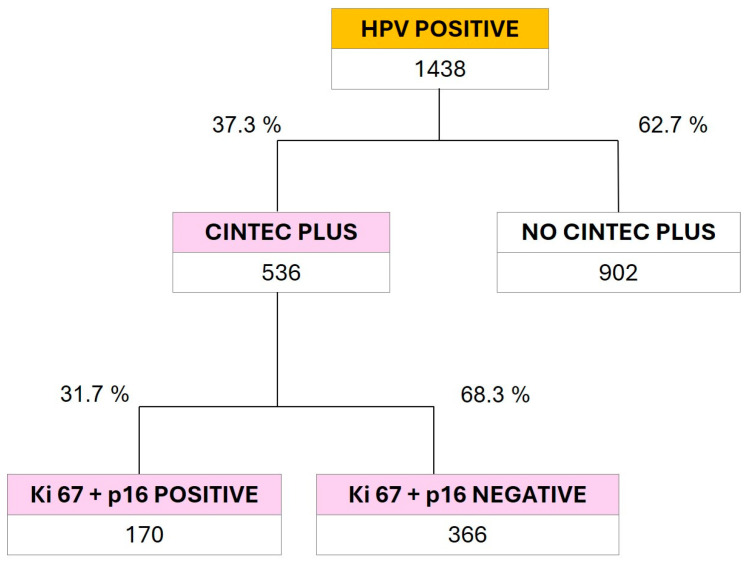
General distribution of CINTEC-PLUS results on samples with positive PCR results for the pool of 12 HPV subtypes at HR for CC (HR12).

**Figure 4 viruses-16-00887-f004:**
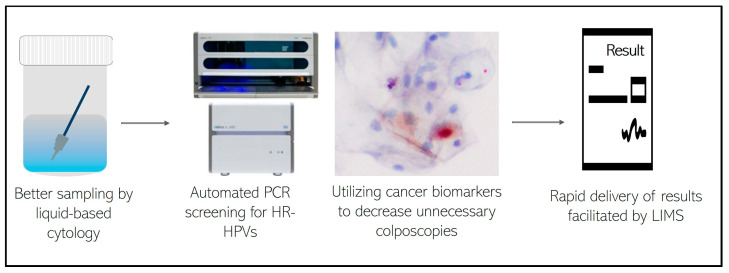
Ideal workflow for the adoption of molecular methods.

**Table 1 viruses-16-00887-t001:** Distribution of the different HPV subtypes by age groups.

	Total PCR	Positivity by Viral Type				
Age (y)		Total	HR-HPV12	HPV16	HPV18	Coinfections
<25	411	181	152	9	1	19
25–35	1724	614	450	57	16	91
>35	1691	409	296	44	12	57
Total	3826	1204	898	110	29	167

HPV: human papillomavirus; HR: high-risk; PCR: polymerase chain reaction; y: years.

**Table 2 viruses-16-00887-t002:** Distribution by age range of the positive PAP results (*n* = 622) with negative HPV PCR results (PCR NEG) versus positive (PCR POS).

	PAP LOW-GRADE		PAP HIGH-GRADE		
AGE (y)	PCR NEG	PCR POS	PCR NEG	PCR POS	Total
<25	10	64	6	10	90
25–35	53	170	21	47	291
>35	45	88	65	43	241
Total	108	322	92	100	622

PCR: polymerase chain reaction; y: years.

**Table 3 viruses-16-00887-t003:** Distribution of the different HPV subtypes in coincident positive PAP and PCR results.

PAP + PCR Positivity	*n* = 422	%
HR12	261	61.8
HPV16	64	15.2
HPV18	11	2.6
HPV16 + HPV18	4	0.9
HR12 + HPV16	67	16.0
HR12 + HPV18	11	2.6
HR12 + HPV16 + HPV18	4	0.9

HPV: human papillomavirus; PCR: polymerase chain reaction; y: years.

## Data Availability

Data are contained within the article.

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
