# Peer review of "Findings and Challenges in Replacing Traditional Uterine Cervical Cancer Diagnosis with Molecular Tools in Private Gynecological Practice in Mexico"

_viruses, 2024, doi:10.3390/v16060887_

Round 1
Reviewer 1 Report
Comments and Suggestions for Authors
Minor editing of English language required
Comments on the Quality of English LanguageThe authors presented the article entitled” Findings and Challenges Replacing the traditional uterine cervical cancer diagnosis with molecular tools in the private gynecological practice in Mexico” by analyzing 4,499 cervix samples. Statistical or Logistic of the results of PCR HPV subtypes, pap and p16/Ki67 dual stain should be added. Limitations of this study such as lack of tissue pathologic study should be described in discussion section.
Author Response
1. Summary |
|
|
Thank you sincerely for dedicating your time to reviewing this manuscript. Enclosed, you will find detailed responses along with corresponding comments, where we have endeavored to explain or indicate the respective changes.
|
||
3. Point-by-point response to Comments and Suggestions for Authors
The authors presented the article entitled” Findings and Challenges Replacing the traditional uterine cervical cancer diagnosis with molecular tools in the private gynecological practice in Mexico” by analyzing 4,499 cervix samples. Statistical or Logistic of the results of PCR HPV subtypes, pap and p16/Ki67 dual stain should be added. Limitations of this study such as lack of tissue pathologic study should be described in discussion section. |
||
Comments 1: Statistical or Logistic of the results of PCR HPV subtypes, pap and p16/Ki67 dual stain should be added |
||
Response 1: We are grateful for your thoughtful and timely feedback on our work. Following a thorough review of the results, we have included a mention at the conclusion of the methodology section (lines 152 to 154) regarding the utilization of contingency tables for data analysis, along with a reference to the software employed. These statistical analyses are subsequently referenced in the results sections (lines 220 to 225) and elaborated upon in the discussion section (line 307 to 310).
|
||
Comments 2: Limitations of this study such as lack of tissue pathologic study should be described in discussion section.
|
||
Response 2: We concur on the significance of acknowledging the study's limitations as an integral part of scientific practice. These limitations have been promptly incorporated into the discussion section (lines 325 to 328)
|
||
4. Response to Comments on the Quality of English Language |
||
Point 1: |
||
Response 1: We reviewed the entire manuscript again and made a couple of changes to improve the English. |

Reviewer 2 Report
Comments and Suggestions for Authors
The authors present the results of a study undertaken to compare molecular tools to traditional tools for cervical cancer identification, specifically, the PCR test and the staining for p16/Ki67 biomarkers. The study sample size seems sufficient for the goal of the study and does show the value of sensitive, specific and less invasive tests to improve cervical cancer diagnosis. My specific comments follow;
1. The results section could be simplified by not repeating in prose what has been shown in the figures and tables, but rather pointing out just the results of note following each figure and table. For example, the lack of correlation between HPV presence and Pap positivity (Figure 2) or a broad summary of the distribution of HPV type by age (Table 1).
2. Given that the second goal of the authors is to encourage higher use of molecular and non-invasive methods, it might be useful to provide a flow chart of how these tools could be used from sample to reporting to a patient and the advantages in terms of accuracy (improvement by X percent), timeliness (improvement by Y days), non invasiveness, and other metrics. Such a chart would be easier to show practitioners and get buy in.
3. Can the authors speculate on why there might be such low correlation between PAP tests and PCR tests? Also, is there another gold standard like sequencing that the results could be compared to.
4. Can the authors comment on any correlations betwr
Author Response
1. Summary |
|
|
We are grateful for the time and dedication you devoted to reviewing our manuscript. We have thoroughly considered each of your comments and will now address them one by one.
|
||
3. Point-by-point response to Comments and Suggestions for Authors
The authors present the results of a study undertaken to compare molecular tools to traditional tools for cervical cancer identification, specifically, the PCR test and the staining for p16/Ki67 biomarkers. The study sample size seems sufficient for the goal of the study and does show the value of sensitive, specific, and less invasive tests to improve cervical cancer diagnosis. My specific comments follow:
|
||
Comments 1: 1. The results section could be simplified by not repeating in prose what has been shown in the figures and tables, but rather pointing out just the results of note following each figure and table. For example, the lack of correlation between HPV presence and Pap positivity (Figure 2) or a broad summary of the distribution of HPV type by age (Table 1). |
||
Response 1: We accurately condensed the description of the results to provide only a summary and highlight the most relevant findings (lines 173 to 183 and lines 220 to 230).
|
||
Comments 2: 2. Given that the second goal of the authors is to encourage higher use of molecular and non-invasive methods, it might be useful to provide a flow chart of how these tools could be used from sample to reporting to a patient and the advantages in terms of accuracy (improvement by X percent), timeliness (improvement by Y days), non invasiveness, and other metrics. Such a chart would be easier to show practitioners and get buy in. |
||
Response 2: We added a diagram (Figure 4) and explained it in the results section (lines 269 to 271). |
Comments 3: Can the authors speculate on why there might be such low correlation between PAP tests and PCR tests? Also, is there another gold standard like sequencing that the results could be compared to. |
Response 3.1: In response to the first question, we corrected the term "low correlation" because what we did was arrange the data in a contingency table to detect false positives and false negatives. What we did was to compare the results of PCR as the standard method with the PAP results. This comparison was added in the results section (lines 221 to 230) and discussed in the corresponding section (lines 307 to 310).
Response 3.2: To address the reviewer's timely observations regarding the second question, we added in the discussion section (lines 324 to 330).
|
Comments 4: Can the authors comment on any correlations betwr |
Response 4: Statistical analyses were added throughout the paper to compare the PCR vs. PAP results previously described (lines 221 to 224).
|
